# Human papillomavirus (HPV) vaccine uptake and its associated factors among adolescent girls in Kathmandu District, Nepal: A cross-sectional study

Shristi Thapa[1]*, Priyanka Timsina[1], Bandana Paneru[1], Archana Pokhrel[1], Archana Shrestha[1,2]

**1** Department of Public Health and Community Programs, Kathmandu University School of Medical Sciences, Dhulikhel, Nepal, **2** Institute for Implementation Science and Health, Kathmandu, Nepal

* thapashristi92@gmail.com

## Abstract

HPV causes over 95% of cervical cancer globally. In Nepal, cervical cancer is the second most common cancer with a crude incidence of 14.2 per 100,000 women, and HPV vaccination for 14-year-old girls began nationwide in 2023. This study assessed HPV vaccine uptake and its associated factors among adolescent girls in Kathmandu district. A cross-sectional study was conducted using multistage cluster random sampling. From 770 schools across eight municipalities where the vaccine available, 142 were selected, with ten 14–15-year-old girls each school. Data were collected using a supervised self-administered questionnaire. HPV vaccine uptake was defined as receipt of two doses within six months. Factors associated with uptake were analysed using Poisson regression with generalized estimating equations, accounting for clustering and sample weights. The prevalence of HPV vaccine uptake was 12.74% (CI: 7.2–21.4). Girls attending private schools had lower uptake than those in public schools (APR: 0.05; CI: 0.02–0.13), while 15-year-old girls had higher uptake than 14-year-olds (APR: 2.28; CI: 1.49–3.50). Lower uptake was observed among girls whose fathers were daily wage labour (APR: 0.59; CI: 0.39–0.90), self-employed (APR: 0.59; CI: 0.39–0.90), or employed abroad (APR: 0.65; CI: 0.43–0.97), whereas higher uptake was noted among girls whose mothers were employed in government jobs (APR: 2.70; CI: 1.55–4.69), private jobs (APR: 2.15; CI: 1.11–4.14), or abroad (APR: 2.58; CI: 1.31–3.86). Good knowledge of HPV infection (APR: 1.88; CI: 1.13–3.15) and good or moderate knowledge of the HPV vaccine (APR: 2.73; CI: 1.33–5.60 and APR: 2.02; CI: 1.07–3.82) were also associated with higher uptake. In conclusion uptake was higher among public school students, older adolescents, and those with working mothers or good HPV knowledge, highlighting the need for targeted interventions to improve awareness, school-based education, and access to vaccination services.

**Data availability statement:** All relevant data underlying the findings of this study are submitted as supplementary information files within this manuscript.

**Funding:** This study was supported by the GTA Foundation Nepal through the Graduate Research Grant 2024 (to ST). The funder had no role in study design, data collection and analysis, decision to publish, or preparation of the manuscript. No authors received a salary from the funder.

**Competing interests:** The authors have declared that no competing interests exist.

## Introduction

Cervical cancer defined as a malignant tumor of the cervix primarily caused by persistent infection with the Human Papillomavirus (HPV) is a major public health problem globally ranking as the fourth leading cause of cancer death among women, with 662,301 new cases and 348,874 deaths of women in 2022 [1]. More than 95% of cervical cancer cases are caused by infection with the Human Papillomavirus (HPV), which is the most common sexually transmitted infection worldwide [2]. About 90% of deaths caused by cervical cancer occurred in low and middle-income countries (LMIC) [3]. South-East Asia has the highest incidence and mortality rates of cervical cancer [3], with approximately 69,000 new cases and 38,000 deaths reported in 2020 [4]. In Nepal, cervical cancer is the most common cancer among women, with a crude incidence rate of 14.2 per 100,000 women and an estimated 2,244 new cases and 1,493 deaths occurring annually [5]. Cervical cancer ranks as the first leading cause of female cancer and the second most common cancer in women aged 15–44 years in Nepal [6].

The World Health Organization recommends two doses of the HPV vaccine for 9–14-year-old girls, which will act as a primary prevention against cervical cancer if it is administered before being sexually active [7]. HPV immunization could control nearly 70% of all cervical cancers [8,9]. HPV vaccines are available in at least 124 countries, including Nepal [10]. Sixty-four countries have launched national HPV immunization programs to achieve the 2030 Sustainable Development Goal (SDG) of vaccinating 90% of girls by age 15 [11]. However, global coverage with the first dose of HPV among girls is now estimated at 21%. [3] Nepal does not have a national HPV immunization program, and no national-level vaccination coverage data is available [7].

In Nepal, there was a pilot project to vaccinate (11–13) year-old girls in two districts from 2016 to 2017 with two doses of Cervarix. However, escalating it and incorporating the program into the regular vaccination program was not achieved [7]. In 2018, a study conducted among adolescent girls in Nepal revealed that only 13.9% of school-going girls were aware of the HPV vaccine. In 2023, the Ministry of Health and Population launched a broader HPV vaccination campaign, purchasing 20,000 HPV vaccines and distributing them through seven major hospitals nationwide. The campaign targeted 14-year-old school-going adolescent girls, administering two vaccine doses over a six-month period [12,13]. But the vaccine coverage is low. For example, only around 600 girls in Kathmandu district have been inoculated with the First dose of the HPV vaccine, although health authorities have targeted to vaccinate 3000 girls [12]. The Family Welfare Division, Department of Health Services (DoHS), releases a new guideline on HPV Vaccine Demonstration 2024 to provide guidance on the HPV vaccine demonstration program in Nepal and a new guideline on HPV Vaccination Service Operational Guidelines 2024 to provide guidance, operational and procedures on the HPV Vaccination service program in Nepal [14]. Nepal's Ministry of Health and Population (MoHP) has announced plans to launch a nationwide HPV vaccination campaign on February 4, 2025. The campaign targets 10–14-year-old girls, regardless of their school enrollment status [15]. Despite these and future targets from the Nepali government, there is a limited understanding of the factors

affecting vaccination uptake in Nepal's context. Existing studies indicate low awareness and limited access to vaccination services, but evidence from Kathmandu district is scarce. Addressing this knowledge gap is critical to design effective interventions and policies to improve vaccination coverage. Factors must be identified earlier and addressed to run an effective vaccine program and increase vaccine uptake among adolescent girls. Therefore, this study aims to estimate the current prevalence of HPV vaccine uptake in the Kathmandu district and identify factors associated with vaccine uptake.

## Methods

### Study setting and design

This is a cross-sectional study among adolescent school girls aged 14–15 years in eight municipalities of Kathmandu district: Kathmandu Metropolitan City, Kageshwori Manohara, Nagarjun, Dakshinkali, Kirtipur, Gokarneshwor, Budhanilkantha, and Chandragiri. These sites received HPV vaccines through a campaign led by the Ministry of Health and Population (MOHP), which targeted 14-year-old girls in 2023. In 2023, Nepal introduced HPV vaccination through a demonstration campaign in selected districts, with support from Global Alliance for Vaccine initiatives and the World Health Organization. The government planned to gradually integrate the vaccine into the National Immunization Program, officially launching it in February 2025. As of 2024, the vaccine had not yet become part of the nationwide routine immunization schedule but remained available in selected municipalities that conducted demonstration or catch-up campaigns. In the study sites particularly the eight municipalities of Kathmandu school-based programs and municipal health facilities provided the HPV vaccine during the campaign period, making these areas appropriate for assessing vaccine uptake and associated factors.

### Study population

The study population comprises school-going adolescent girls aged 14–15 years who were enrolled in grades 9 and 10 in all eligible secondary-level schools across eight municipalities of Kathmandu district during the study period, as identified from official school lists obtained from the respective municipalities. Inclusion criteria are: school-going adolescent girls aged 14–15 years enrolled in schools within the study site at the time of data collection, aligning with the MOHP's 2023 vaccination initiative targeting school-going 14-year-old girls. We excluded those with hearing or visual impairments to ensure participants could fully comprehend the questionnaire and respond accurately without requiring assistance. This was important to maintain consistency in data collection and avoid introducing bias related to interpretation or support needs during the survey process. Those absent from class during data collection were also excluded.

For participant recruitment, we first obtained permission from the schools. We then screened girls aged 14–15 in grades 9 and 10 to determine their eligibility. Eligible students received a brief explanation of the study and its purpose. We obtained written consent forms from parents and written assent forms from the participants. To ensure informed participation, we sent the consent form to parents via the participants, then we followed up with parents by phone to explain the study, address any questions or concerns, and confirm their willingness to allow their daughters to participate. We emphasized ethical considerations, including the voluntary nature of participation, the right to withdraw at any time, and the assurance of privacy and confidentiality. We also made it clear that there would be no teacher coercion or pressure to participate.

### Ethical statement

The ethical clearance was obtained from the Institutional Review Committee (IRC) of Kathmandu University School of Medical Sciences (IRC-KUSMS Approval NO. 68/24). The study team obtained written informed consent from the respondents' parents and assent from the respondents. Respondents were informed about the research details, including objectives, their role in the study, and the risks and benefits of participation. Participation was voluntary, and withdrawal from the study at any time during data collection was considered. No name or other identifying information was included in the questionnaire.

## Sample Size

The final sample size for the study was 1,560 participants. We first calculated the base sample size using a single population proportion formula with a 95% confidence interval, 5% margin of error, and an estimated HPV vaccine uptake prevalence of 19.6%, based on a study from Uganda [16] a setting similar to Nepal in terms of vaccine accessibility and awareness:

$$n \ = \ p(1-p) \ (z/e)^2 \ = \ 0.196 * (1-0.196) \ * (1.96/0.05)^2 \ = \ 242.14$$

We then adjusted for clustering using a design effect of 5.86. We take an intraclass correlation coefficient (δ) of 0.54 [17] and a cluster size of 10 students per school to calculate the design effect as 1 + ICC(cluster size- 1). Multiplying the base sample by the design effect gave an adjusted sample size of 1,418. Finally, we added a 10% non-response rate with the final sample size of 1560.

## Sampling technique

We conducted a multi-stage cluster random sampling with secondary-level schools as clusters. In Stage 1, we randomly selected 142 schools from 770 schools in 8 municipalities/metropolitan cities, ensuring that the selection was proportionate to the total number of secondary schools and stratified by the private or public nature of the school. If a selected school was unavailable or did not respond, we replaced it with the next school from the same stratum within the same municipality or metropolitan city. In Stage 2, we selected grades 9 and 10 from the schools because the HPV vaccine was administered to 14-year-old girls in these grades. In Stage 3, we used systematic random sampling to select 10 participants in each class by creating a sampling frame of all eligible girls aged 14–15 from all grades 9 and 10 sections according to their class roll numbers. The interval was calculated by dividing the total number of eligible girls by the desired sample size of 10 in each school. A random starting point was selected, and every nth participant was chosen (Fig 1).

## Data collection tools

We developed structured questionnaires in English, translated them into Nepali, and back-translated them into English to ensure consistency. Experts reviewed the content for validity. The questionnaires were adapted from existing literature and previous studies. [18] The tool included five sections assessing: (1) socio-demographics; (2) knowledge of cervical cancer, HPV infection, and vaccination; (3) perceptions of HPV vaccination and related diseases; (4) promotional exposure; and (5) information sources on HPV vaccination and uptake. We collected data from 15th July to 15th September 2024 using self-administered questionnaires in Nepali language.

## Measurements

**Socio-demographics variables.** The socio-demographic variables included Age (in years), Education (number of years of formal education), Ethnicity (Adivasi/janajati, brahmin/Chhetri, newar, dalit, terai/Madhesi and others), Religion (Hinduism, Buddhism, Christianity, Muslim and others), Mother's and fathers education status (number of years of formal education), Family Type (Nuclear (parents and their children only), Joint (consisting of grandparents, great-grandparents, paternal grandparents, cousins and their children), Number of family members, Marital status of parents (married, separated, and widow), Media Sources (radio, television, internet and newspaper, others), Mother's and fathers employment status (full-time job, part-time job and unemployed), (Mother's and Father's) occupation (agriculture, building construction, daily wages, government job, private job, business, home-maker, foreign employment, others) and Annual family income in Nepalese currency.

**Knowledge-related variables.** We measured knowledge about cervical cancer using a set of six questions, knowledge about HPV infection utilizing a set of five questions, and knowledge about the HPV vaccine using a set of

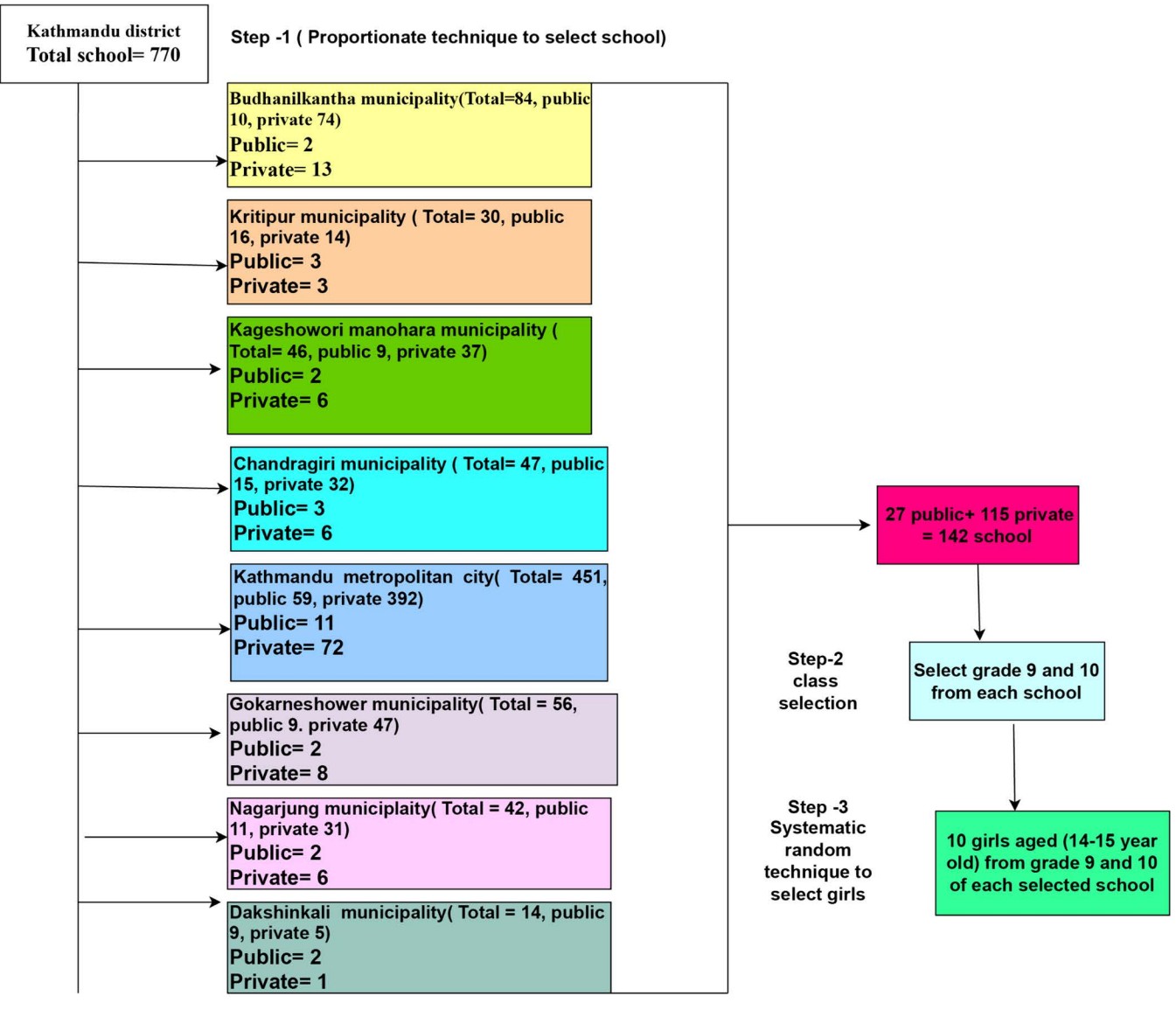

**Fig 1. Sampling frame of study.**

nine questions. We awarded a score of 1 for each correct response and 0 for each incorrect response, then translated it into a percentage of correct responses, indicating higher scores and better knowledge. The same approach as in other studies was used to measure knowledge levels. [19,20] Participants who scored above 80% were classified as having 'good knowledge,' scoring 50–79% as having 'moderate knowledge,' and those scoring below 50% as having 'poor knowledge' [20].

**Perceptions-related variables.** We measured perceptions towards HPV vaccination using 30 perception-related questions based on six categories of the Health Belief Model: Susceptibility, Severity, Benefits/Facilitators, Barriers, Self-efficacy, and Cues to Action, on a five-point Likert scale. We classified participants with scores above the mean as having a 'positive perception' towards HPV vaccination and those below the mean as having a 'negative perception.' The same approach is used in other studies to measure perception [21].

**HPV vaccine uptake-related information.** In this study, uptake of the HPV vaccine was defined as the proportion of those 14–15-year-old girls who had received two doses of the HPV vaccine within a six-month interval. We measured vaccine uptake using full vaccination status, which is defined as receiving two doses of the HPV vaccine within a six-month interval. Any adolescent who had received one or no dose of the HPV vaccine was considered not fully vaccinated. HPV vaccine uptake was determined using respondents' recall or vaccination cards. Adolescents were considered fully vaccinated if they self-reported to have received two doses of the HPV vaccine within a six-month interval. We cross-verified responses with vaccination cards and triangulated data with school records to minimize recall bias. Since the second dose of the vaccine was administered recently, in May 2024, we conducted data collection between July and September, ensuring a short recall period.

## Data quality control

To maintain the quality of the data, data collectors were trained in data collection procedures before the actual data collection time; the questionnaire (tool) was pretested for validity and reliability on 10% of the total sample size in a similar setting in Bhaktapur district to assess instrument simplicity, flow, and consistency; thereby possible adjustment or modification was considered/ on a time of deliverance and minor correction on some questions had been incorporated. We checked the internal consistency of each section using Cronbach's alpha test for knowledge about cervical cancer (alpha value = 0.72), knowledge about HPV infection (alpha value = 0.98), knowledge about HPV vaccine(alpha value = 0.84) and perception about cervical cancer, HPV infection and HPV vaccine(alpha value = 0.84) and overall total Cronbach alpha value is 0.87. The principal investigator and supervisor spot-checked and reviewed all the completed questionnaires to ensure completeness and consistency of the information collected. A principal investigator supervised the data collectors. The principal investigator has entered the data to ensure its accuracy into the Kobo Collect app daily.

## Data analysis

We calculated frequencies and percentages for categorical variables and means and standard deviations for continuous variables, adjusting for the individual sampling weights of girls. We conducted generalized estimating equations using bivariate and multivariate Poisson regression models, an exchangeable working correlation, and robust variance after accounting for cluster sampling by school and adjusting for individual sample weights of the girls to assess the association between socio-demographic characteristics, knowledge, and perception as exposures and HPV vaccine uptake as the outcome. Variables for the multivariable model were selected based on prior literature and theoretical relevance. We reported crude (CPR) and adjusted (APR) prevalence ratio at a 95% confidence interval (CI) and statistically significant if the p-value< 0.05. All the data analysis was performed using STATA software version 14.

## Results

Table 1 represents the Socio-demographic characteristics of respondents. A total of 1510 girls participated in the study, with a response rate of 96.79%. Respondents had a mean age of 14.63 (SD = 0.48) years. The majority 47.14% were from Brahmin/Chettri ethnicity and followed Hinduism. Regarding parental educational status majority,41.64% of mothers and 48.97% of fathers had completed secondary education. Most families were nuclear (81.81%), with an average family size of 5.08 members and 2.28 siblings per family. Most respondents (92.39%) had married parents, and 91.35% reported using the Internet as their primary media source. The annual per capita income averaged USD 1579.22, with a median of USD 980.39 and an Interquartile range of USD 1225.49 (Table 1).

Majority (57.38%) of the respondents had poor knowledge about cervical cancer, (79.72%) had poor knowledge about HPV infection, and most of respondents 79.36% had poor knowledge about the HPV vaccine. Regarding perceptions about cervical cancer, HPV infection, and the HPV vaccine, while majority 54.51% had positive perceptions and 45.49% had negative perceptions (Table 2).

**Table 1. Socio-demographic characteristics of respondents (n = 1510).**

| Variables | Weighted Percent (%) | Weighted frequency | Unweighted Frequency |
|---|---|---|---|
| **School type** | | | |
| Government | 32.57 | 492 | 321 |
| Private | 67.43 | 1018 | 1189 |
| **Class** | | | |
| Nine | 51.94 | 784 | 792 |
| Ten | 48.06 | 726 | 718 |
| **Age, in years (Mean±SD)** | | (14.63±0.482) | 14.62±0.48 |
| **Ethnicity** | | | |
| Adibasi/janajati | 42.07 | 635 | 633 |
| Brahmin/Chettri | 47.14 | 712 | 711 |
| Terai/Madhesi | 6.38 | 96 | 91 |
| Dalit and others | 4.41 | 67 | 75 |
| **Religion** | | | |
| Hindu | 77.65 | 1173 | 1197 |
| Non-Hindu | 22.35 | 337 | 313 |
| **Mother's education** | | | |
| No formal education | 19.37 | 292 | 273 |
| Basic education (1–8) | 29.23 | 441 | 437 |
| Secondary (9–12) | 41.64 | 629 | 643 |
| 12+ and above | 9.76 | 147 | 157 |
| **Mother's mean years of schooling (Mean±SD)** | | 7.678(5.129) | 7.81±5.13 |
| **Father's education** | | | |
| No formal education | 13.18 | 199 | 163 |
| Basic education (1–8) | 20.94 | 316 | 315 |
| Secondary (9–12) | 48.97 | 739 | 761 |
| 12+ and above | 16.92 | 255 | 271 |
| **Father's mean years of schooling (Mean±SD)** | | (9.340±5.183) | (9.60±5.00) |
| **Family Type** | | | |
| Nuclear | 81.81 | 1235 | 1,229 |
| Joint | 18.19 | 275 | 281 |
| **Number of Family members (n, Mean±SD)** | | (5.081±2.162) | 1510 (5.03±2.07) |
| **Number of siblings (n, Mean±SD)** | | (2.28±1.157) | 1,510, (2.24±1.12) |
| **Parents marital status** | | | |
| Married | 92.39 | 1395 | 1,396 |
| Separated | 4.74 | 72 | 71 |
| Widow | 2.86 | 43 | 43 |
| **Most used media sources\*** | | | |
| TV | 10.19 | 154 | 140 |
| Internet | 91.35 | 1379 | 1389 |
| Others(radio, newspaper) | 5.25 | 79 | 79 |
| **Fathers' employment status** | | | |
| Full-time job | 87.58 | 1284 | 1,304 |
| Part-time job | 8.33 | 122 | 111 |
| Unemployed | 4.09 | 60 | 51 |
| **Fathers' occupation** | | | |
| Agriculture | 7.26 | 107 | 81 |

*(Continued)*

**Table 1.** (Continued)

| Variables | Weighted Percent (%) | Weighted frequency | Unweighted Frequency |
|---|---|---|---|
| Daily waged labor | 9.60 | 141 | 143 |
| Government job | 11.98 | 177 | 160 |
| Private job | 16.98 | 250 | 269 |
| Business | 22.00 | 324 | 343 |
| Homemaker | 2.29 | 34 | 35 |
| Foreign Employment | 14.86 | 219 | 235 |
| Others(carpet weaver, driver) | 15.04 | 222 | 208 |
| **Mothers' employment status** | | | |
| Full-time job | 48.86 | 730 | 749 |
| Part-time job | 13.90 | 208 | 210 |
| Unemployed | 37.25 | 557 | 536 |
| **Mothers' occupation** | | | |
| Agriculture | 5.77 | 87 | 75 |
| Daily waged labor | 9.19 | 138 | 140 |
| Government job | 5.76 | 87 | 80 |
| Private job | 12.46 | 187 | 205 |
| Business(self-employed) | 17.60 | 265 | 285 |
| Homemaker | 39.18 | 589 | 569 |
| Foreign Employment | 3.21 | 48 | 47 |
| Others(house helper, carpet weaver) | 6.84 | 103 | 102 |
| **Earns money in family*** | | | |
| Father | 38.30 | 578 | 542 |
| Mother | 8.11 | 122 | 95 |
| Both father and mother | 52.35 | 791 | 840 |
| None | 1.09 | 16 | 17 |
| **Father's annual income NRs (Mean± SD, MD, IQR)** | (594012.6±1096113) (360000 600000) | | (617474.9±1114296) (420000,600000) |
| **Mothers annual income NRs (Mean±SD, MD, IQR)** | (227913.5±1244592) MD: 114000 IQR: 300000 | | (231848.9±1274016) MD:114000 IQR:300000 |
| **Annual per capita income, in USD (Mean±SD, MD, IQR)** | 1579.22±4123.73 MD: 980.3922 IQR: 1225.49 | | 1630.54±4274.11 MD: 1069.519 IQR:1247.772 |

*Multiple responses, MD: Median, IQR: Interquartile range, SD: Standard Deviation

Among the 1510 respondents, only 12.70% (95% CI: 7.2–21.4) were fully vaccinated, with two doses over a six-month interval and 1.2% were partially vaccinated with one dose (Fig 2).

Table 3 presents the factors associated with HPV vaccine uptake. In the bivariate model, respondents studying in private schools had a 96% lower prevalence of receiving the HPV vaccine compared to those in public schools (CPR: 0.04; 95% CI: 0.01–0.12; p<0.0001). Respondents aged 15 had 95% higher prevalence of receiving the HPV vaccine compared to those aged 14 years (CPR: 1.95; 95% CI: 1.23–3.08; p=0.003). Respondents whose fathers were self-employed in business had a 26% lower prevalence of receiving the HPV vaccine compared to those whose fathers worked in agriculture (CPR: 0.74; 95% CI: 0.56–0.99; p=0.04). Respondents whose mothers held government jobs had a 53% higher prevalence of receiving the HPV vaccine compared to those whose mothers were homemakers (CPR: 1.53;

**Table 2. Knowledge and perception Level of respondents about Cervical cancer, HPV infection, and HPV vaccine.**

| Level of knowledge | Weighted percent% | Weighted frequency | Unweighted frequency |
|---|---|---|---|
| **Knowledge about Cervical cancer** | | | |
| Poor knowledge | 57.38 | 866 | 960 |
| Moderate Knowledge | 19.57 | 295 | 191 |
| Good knowledge | 23.05 | 384 | 359 |
| Mean (SD) | | 33.28 (39.51) | 30.03(40.3) |
| **Knowledge about HPV infection** | | | |
| Poor knowledge | 79.72 | 1204 | 1227 |
| Moderate knowledge | 3.08 | 46 | 29 |
| Good knowledge | 17.20 | 260 | 254 |
| Mean (SD) | | 18.53(35.87) | 17.44(35.76) |
| **Knowledge about HPV vaccine** | | | |
| Poor knowledge | 79.36 | 1198 | 1269 |
| Moderate Knowledge | 13.13 | 198 | 144 |
| Good Knowledge | 7.51 | 113 | 97 |
| Mean (SD) | | 17.73(31.79) | 13.70(29.36) |
| **Perception towards cervical cancer, HPV infection, and HPV vaccination** | | | |
| Negative Perception | 45.49 | 687 | 733 |
| Positive perception | 54.51 | 823 | 777 |
| Mean (SD) | | 101.49 (14.164) | 100.60(12.93) |

*A score above 80% is considered as 'good knowledge', 50–79% as 'moderate knowledge' and below 50% as 'poor knowledge'

* A score above the mean is classified as having a 'positive perception,' and below the mean is classified as a 'negative perception.'

## Prevalence of HPV vaccine uptake (n=1510)

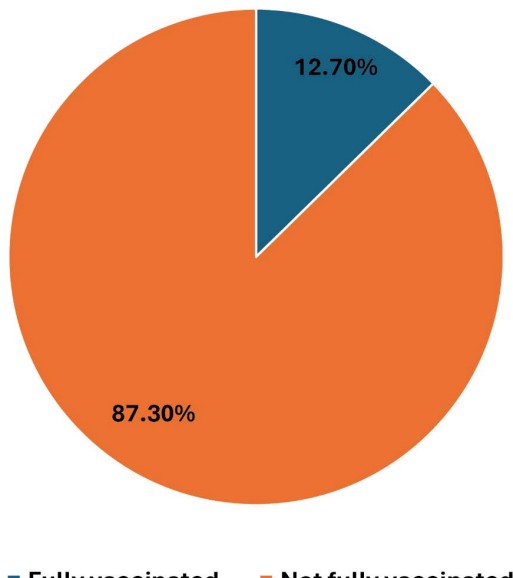

■ **Fully vaccinated**   ■ **Not fully vaccinated**

**Fig 2. Prevalence of HPV vaccine uptake.**

**Table 3. Association of HPV vaccine uptake with socio-demographic factors(n = 1510).**

| Variables | Bivariate | | | Multivariate | | |
|---|---|---|---|---|---|---|
| | CPR | 95% CIs | P-value | APR | 95% CI | P-value |
| **School type** | | | | | | |
| Public | Ref | | | | | |
| Private | 0.04 | 0.01,0.12 | <0.0001 | 0.05 | 0.02,0.13 | <0.0001* |
| **Age** | | | | | | |
| 14 | Ref | | | | | |
| 15 | 1.95 | 1.23,3.08 | 0.003 | 2.28 | 1.49,3.50 | <0.0001* |
| **Ethnicity** | | | | | | |
| Adibasi/Janjati | Ref | | | | | |
| Brahmin/Chettri | 0.77 | 0.55,1.07 | 0.13 | 0.77 | 0.58,1.02 | 0.07 |
| Terai/Madhesi | 0.86 | 0.63,1.17 | 0.35 | 0.60 | 0.34,1.06 | 0.08 |
| Dalit and others | 0.95 | 0.57,1.59 | 0.85 | 0.94 | 0.66,1.49 | 0.97 |
| **Religion** | | | | | | |
| Hindu | Ref | | | | | |
| Non-Hindu | 1.25 | 0.88,1.76 | 0.20 | 1.10 | 0.78,1.56 | 0.56 |
| **Mothers' education** | | | | | | |
| No formal education | Ref | | | | | |
| Basic education (1–8) | 0.97 | 0.78,1.19 | 0.77 | 0.90 | 0.72,1.12 | 0.35 |
| Secondary (9–12) | 0.90 | 0.64,1.27 | 0.56 | 0.93 | 0.66,1.33 | 0.72 |
| 12+ and above | 0.95 | 0.59,1.52 | 0.83 | 1.28 | 0.48,3.40 | 0.61 |
| **Fathers' education** | | | | | | |
| No formal education | Ref | | | | | |
| Basic education (1–8) | 1.03 | 0.82,1.31 | 0.74 | 1.13 | 0.92,1.38 | 0.22 |
| Secondary (9–12) | 0.86 | 0.71,1.06 | 0.16 | 1.02 | 0.78,1.27 | 0.99 |
| 12+and above | 0.84 | 0.56,1.25 | 0.39 | 0.87 | 0.45,1.67 | 0.69 |
| **Family Type** | | | | | | |
| Nuclear | Ref | | | | | |
| Joint | 1.10 | 0.90,1.36 | 0.30 | 1.25 | 0.76,1.38 | 0.37 |
| **Total family members** | 1.01 | 0.96,1.06 | 0.58 | 0.95 | 0.88,1.02 | 0.21 |
| **Total siblings** | 1.02 | 0.85,1.22 | 0.81 | 1.04 | 0.91,1.19 | 0.50 |
| **Parents marital status** | | | | | | |
| Married | Ref | | | | | |
| Separated | 1.04 | 0.76,1.44 | 0.77 | 0.64 | 0.40,.03 | 0.06 |
| Widow | 1.18 | 0.93,1.51 | 0.16 | 1.21 | 0.76,1.92 | 0.42 |
| **Fathers' occupation** | | | | | | |
| Agriculture | Ref | | | | | |
| Daily wages | 0.89 | 0.63,1.25 | 0.50 | 0.59 | 0.39,0.90 | 0.015* |
| Government job | 0.94 | 0.85,1.54 | 0.83 | 0.83 | 0.48,1.44 | 0.53 |
| Private job | 0.61 | 0.36,1.03 | 0.06 | 0.51 | 0.25,1.05 | 0.07 |
| Business | 0.74 | 0.56,0.99 | 0.04 | 0.65 | 0.43,0.97 | 0.03* |
| Homemaker | 0.98 | 0.62,1.55 | 0.96 | 1.09 | 0.56,2.13 | 0.79 |
| Foreign Employment | 0.67 | 0.45,1.02 | 0.06 | 0.61 | 0.38,0.98 | 0.04* |
| Others(carpet weaver, driver) | 0.75 | 0.47,1.21 | 0.25 | 0.62 | 0.39,1.01 | 0.05 |

*(Continued)*

| Variables | Bivariate | | | Multivariate | | |
|---|---|---|---|---|---|---|
| | CPR | 95% CIs | P-value | APR | 95% CI | P-value |
| **Mothers' occupation** | | | | | | |
| Homemaker | Ref | | | | | |
| Daily wages | 1.08 | 0.70,1.66 | 0.71 | 1.41 | 0.87,2.54 | 0.14 |
| Government job | 1.53 | 1.05,2.25 | 0.02 | 2.70 | 1.55,4.69 | <0.0001* |
| Private job | 1.29 | 0.98,1.70 | 0.06 | 2.15 | 1.11,4.14 | 0.02* |
| Business | 0.70 | 0.41,1.19 | 0.19 | 0.93 | 0.52,1.86 | 0.97 |
| Agriculture | 0.97 | 0.59,1.58 | 0.90 | 1.27 | 0.73,2.22 | 0.38 |
| Foreign Employment | 1.68 | 0.96,2.92 | 0.06 | 2.58 | 1.31,3.86 | 0.003* |
| Others(house helper, carpet weaver) | 0.77 | 0.4,1.33 | 0.36 | 0.97 | 0.50,1.94 | 0.97 |
| **Earns money for the family** | | | | | | |
| Father | Ref | | | | | |
| Mother | 0.97 | 0.60,1.56 | 0.91 | 0.55 | 0.31,0.97 | 0.03* |
| Both father and mother | 1.05 | 0.81,1.36 | 0.68 | 0.81 | 0.52,1.26 | 0.35 |
| None | 1.99 | 0.92,4.30 | 0.07 | 1.32 | 0.56,3.09 | 0.52 |
| Others family members | 0.70 | 0.42,1.17 | 0.18 | 0.52 | 0.26,1.05 | 0.06 |
| **Household Annual per capita income (USD)** | 0.99 | 0.99,0.99 | 0.01 | 0.99 | 0.99,1 | 0.41 |

\* CPR: Crude prevalence ratio, APR: Adjusted prevalence ratio

95% CI: 1.05–2.25; p = 0.02). A higher household annual per capita income significantly lowered the HPV vaccine prevalence (CPR: 0.99; 95% CI: 0.99–0.99; p = 0.01). In the multivariate model, respondents studying in private schools had a 95% lower prevalence of vaccine uptake compared to those in public schools (APR = 0.05; 95% CI: 0.02–0.13; p < 0.0001). Respondents aged 15 were 2.28 times more likely to be vaccinated than those aged 14 (APR = 2.28; 95% CI: 1.49–3.50; p < 0.0001). Compared to fathers in agriculture, daughters of daily wage workers, self-employed, and migrant workers had significantly lower prevalence of vaccine uptake—by 41% (APR = 0.59; p = 0.015), 35% (APR = 0.65; p = 0.03), and 39% (APR = 0.61; p = 0.04), respectively. Respondents whose mothers held government (APR = 2.70; p < 0.0001), private (APR = 2.15; p = 0.02), or foreign employment (APR = 2.58; p = 0.003) jobs were significantly more likely to be vaccinated than those with homemaker mothers. Respondents from households where mothers were the sole earners had 45% lower uptake compared to those where fathers were sole earners (APR = 0.55; 95% CI: 0.31–0.97; p = 0.03). All analysis were accounted for school level clustering. No significant associations were observed for ethnicity, religion, parental education, family type, number of siblings, parents' marital status, or household income.

Table 4 presents the association between HPV vaccine uptake and respondents' knowledge and perception levels. In the bivariate model respondents with good knowledge of cervical cancer exhibited a 2.03 times higher prevalence of HPV vaccine uptake compared to those with poor knowledge (CPR: 2.03; 95% CI: 1.24–3.34; p = 0.005)., adolescent girls with good knowledge of HPV infection had a 2.72 times higher prevalence of receiving the HPV vaccine compared to those with poor knowledge (CPR: 2.72; 95% CI: 1.51–4.88; p = 0.001). Participants with good knowledge of the HPV vaccine had a 3.91 times higher prevalence of receiving the vaccine (CPR: 3.91; 95% CI: 1.51–10.11; p = 0.005), while those with moderate knowledge showed a 2.73 times higher prevalence (CPR: 2.73; 95% CI: 1.31–5.71; p = 0.007) compared to those with poor knowledge. In the multivariate model, after adjusting for covariates and accounting for school-level clustering, respondents with good knowledge of HPV infection were 1.88 times more likely to receive the vaccine than those with poor knowledge (APR = 1.88; 95% CI: 1.13–3.15; p = 0.01). Similarly, respondents with good and moderate knowledge of

**Table 4.  Association of HPV vaccine uptake with knowledge and perception level of the respondents(n = 1510).**

| Categories | Bivariate | | | Multivariate | | |
|---|---|---|---|---|---|---|
| | CPR | 95% CI | P-value | APR | 95% CI | P-value |
| **Level of knowledge about cervical cancer** | | | | | | |
| Poor knowledge | Ref | | | | | |
| Moderate knowledge | 1.52 | 0.76,3.04 | 0.23 | 1.44 | 0.67,3.05 | 0.34 |
| Good Knowledge | 2.03 | 1.24,3.34 | 0.005 | 1.59 | 0.99,2.56 | 0.05 |
| **Level of knowledge about HPV infection** | | | | | | |
| Poor knowledge | Ref | | | | | |
| Moderate knowledge | 1.57 | 0.73,3.37 | 0.23 | 1.76 | 0.95,3.27 | 0.07 |
| Good Knowledge | 2.72 | 1.51,4.88 | 0.001 | 1.88 | 1.13,3.15 | 0.01* |
| **Level of knowledge about the HPV vaccine** | | | | | | |
| Poor knowledge | Ref | | | | | |
| Moderate knowledge | 2.73 | 1.31,5.71 | 0.007 | 2.02 | 1.07,3.82 | 0.029* |
| Good Knowledge | 3.91 | 1.51,10.11 | 0.005 | 2.73 | 1.33,5.60 | 0.006* |
| **Perception level of the participants** | | | | | | |
| Negative perception | Ref | | | | | |
| Positive perception | 1.33 | 0.88,2.02 | 0.17 | 1.31 | 0.88,1.95 | 0.18 |

* CR: Crude prevalence ratio, APR: Adjusted prevalence ratio.

**Adjusting for age, ethnicity, religion, parents' marital status, family type, mothers' education, fathers' education, fathers' occupation, mothers' occupation, annual per capita income, initiatives or campaigns promoting HPV vaccination awareness, taught about sexually transmitted disease in school, taught about HPV or cervical cancer at school, school provided any information about the HPV vaccine, health workers inform/teach about cervical cancer or HPV.

the HPV vaccine had 2.73 (APR = 2.73; 95% CI: 1.33–5.60; *p* = 0.006) and 2.02 (APR = 2.02; 95% CI: 1.07–3.82; *p* = 0.029) times higher prevalence of HPV uptake, respectively, compared to those with poor knowledge.

## Discussion

In this study, only about 12.7% of adolescent girls in Kathmandu District had received the HPV vaccine, indicating that uptake remains quite low. Most of the respondents did not know much about cervical cancer (57%), HPV infection (80%), or the HPV vaccine (79%). Approximately 46% of respondents had a negative perception about the vaccine. However, (79%) of respondents who had not been vaccinated said they were willing to get the vaccine. Several factors were linked to HPV vaccination. Respondents who were 15 years old, attended public schools, came from higher-income families, or had parents with formal jobs were more likely to be vaccinated. Those who had better knowledge about HPV and the vaccine were also more likely to have received it.

Our study found that HPV vaccination uptake among adolescent girls in Kathmandu Valley was 12.7%, lower than the 90% coverage reported during pilot vaccination in Nepal's Chitwan and Kaski districts [7]. The pilot achieved high coverage which might be due to its controlled setting, close monitoring, and targeted delivery to over 14000 girls aged 11 and 13, and 10-year-olds out of school at health facilities [7]. Compared to neighbouring and high-income countries, HPV vaccine uptake in our study remains substantially lower as the prevalence of HPV vaccine uptake is lower than the study conducted in India (74.4%), Uttar Pradesh (UP) (79.17%), China (66.9%) [22–24], Taiwan (91%) and in Scotland (94.4%) [25]. These countries benefit from well-established healthcare systems and national immunization programs that support broader vaccine access [22–24]. In Nepal, the government's 2023 demonstration campaign was limited in scope, targeting only 14-year-old schoolgirls with 20,000 doses at the time of the study. In contrast, Nepal's uptake closely resembles rates reported in Uganda (8.6%) [20] and Ibadan, Nigeria (4.1%) [26]. Across these settings, studies consistently identify limited

awareness, misinformation, cultural stigma surrounding adolescent reproductive health, and inconsistent vaccine supply as key barriers to vaccination [20,26]. Additionally, lack of information regarding vaccine availability emerged as a common reason for non-vaccination, aligning with findings from Ethiopia and Uganda [20].

We found that adolescent girls attending public schools and those aged 15 years were significantly more likely to be vaccinated than their counterparts. This pattern reflects Nepal's campaign eligibility criteria, which primarily targeted 14-year-old girls enrolled in public schools, thereby excluding private school students and those outside the specified age range [12,13]. These findings underscore how policy design and implementation strategies directly shape vaccine uptake and may inadvertently create inequities.

Parental occupation also played an important role. Respondents whose fathers were daily wage workers, self-employed, or employed abroad had 41%, 35%, and 39% lower prevalence of HPV vaccine uptake, respectively, compared to those whose fathers worked in agriculture. This contrasts with studies from Ethiopia and Uganda, where fathers' occupation was not significantly associated with vaccine uptake among adolescent girls [20,27]. This may be because fathers engaged in daily wage work, self-employment, or foreign employment often face barriers such as irregular work hours, limited access to health information, and reduced involvement in their children's health decisions due to demanding work schedules. In contrast, adolescent girls whose mothers held government jobs were 2.7 times more likely to receive the HPV vaccine, those with mothers in private jobs were 2.15 times more likely, and those whose mothers worked abroad were 2.58 times more likely, compared to girls whose mothers were homemakers. These findings align with a study in Ethiopia, which also found a significant association between mothers' occupation and HPV vaccine uptake [18]. A study conducted in Kampala, Uganda, found that adolescent girls whose mothers worked in healthcare had a 1.94 times higher likelihood of being fully vaccinated against HPV [20]. Similarly, in Nepal, mothers employed in government or private sectors or working abroad may have better access to health information, greater exposure to health campaigns, and more financial autonomy, enabling them to make informed decisions about their children's health.

We also observed 45% lower HPV vaccine uptake among girls from households where mothers were the sole earners compared to those where fathers were the sole earners. This finding differs from studies in Uganda and India, which reported no significant association between household income source and vaccine uptake [18,22]. In Nepal, patriarchal norms often grant male earners more decision-making power over healthcare. Sole-earning mothers may face challenges balancing work and caregiving, limiting their ability to prioritize preventive health. Additionally, stigma and financial instability in women-headed households may further restrict access to vaccination services [7].

In our study, religion, ethnicity, and parental education were not significantly associated with HPV vaccine uptake, unlike findings from India, the U.S, and Malaysia [28,29]. This may be due to Nepal's 2023 HPV vaccination campaign, which targeted public schools uniformly, minimizing disparities in access across religious and ethnic groups [12,13]. Moreover, the campaign's school-based approach likely reduced the influence of parental education by ensuring equal access to information and vaccination for all eligible students.

Knowledge-related factors played a critical role in vaccine uptake. In our study, knowledge of cervical cancer was associated with HPV vaccine uptake in the bivariate analysis but not in the multivariate model. This contrasts with findings from Ethiopia, where girls with good cervical cancer knowledge were 2–2.8 times more likely to be vaccinated [30,31]. In Nepal, awareness of cervical cancer may not directly lead to vaccination if individuals are unaware that the HPV vaccine prevents it. Our findings showed vaccine uptake was more strongly linked to knowledge, specifically about the HPV vaccine. Structural barriers such as limited information, vaccine availability, cost, and access, along with parental influence on health decisions, may further limit uptake despite awareness [5].

Adolescent girls with good knowledge of HPV infection were twice as likely to receive the vaccine, while those with good knowledge of the HPV vaccine had the strongest association—nearly four times higher uptake in crude analysis and 2.74 times higher after adjustment. These findings align with studies from Ethiopia and Tanzania, where vaccine uptake was four times higher among knowledgeable girls [32,33]. Awareness of HPV and its vaccine enhances understanding of

risks, dispels myths, and fosters trust in vaccination, leading adolescents to seek information, make informed choices, and influence parental decisions—ultimately improving uptake [32,33]. In both bivariate and multivariate models, no association was found between perceptions of cervical cancer, HPV infection, or HPV vaccination and vaccine uptake, contrasting with findings from Uganda where positive perceptions significantly increased uptake [20]. In Nepal, this lack of association may stem from barriers beyond perception, such as limited vaccine access, cultural norms, and socio-economic constraints. Moreover, adolescent vaccination decisions often depend on parents or guardians, whose views may differ from the adolescents' own perceptions [7].

To our knowledge, this is the first study in Nepal to examine HPV vaccine uptake and associated factors among adolescent girls. With a large sample of 1,510 girls aged 14–15, the findings are robust and generalizable within similar settings. We used Generalized Estimating Equations to account for clustering by school and applied sample weights, with prevalence ratios providing clear and accurate associations. However, the study has limitations. It was conducted solely in urban Kathmandu, excluding rural areas and out-of-school girls, which may affect broader applicability. Additionally, the focus on 14–15-year-olds limits generalization to the recommended age range of 10–19 years. Finally, certain factors that may influence HPV vaccine uptake: such as vaccine access, parental decision-making, and broader health system influences, were not assessed in this study due to its cross-sectional design and focus on adolescent respondents.

## Conclusion

In conclusion, HPV vaccine uptake among adolescent girls was low. While encouragement from health workers and prior knowledge facilitated vaccination, lack of information remained a major barrier. Despite low coverage, most unvaccinated girls were willing to receive the vaccine. Poor knowledge and negative perceptions about cervical cancer, HPV, and the vaccine were common. Uptake was significantly associated with school type, age, parental occupation, household income, and knowledge levels. To improve coverage, regular vaccination campaigns and targeted education are essential to foster positive attitudes. Future community-based studies are recommended to address current limitations and broaden understanding.

## Supporting information

**S1 File. Data collection tool.**
(DOCX)

**S2 File. Dataset.**
(XLSX)

## Acknowledgments

We thank the Department of Public Health, Kathmandu University School of Medical Sciences. We would also like to thank the data collectors and study participants.

## Author contributions

**Conceptualization:** Shristi Thapa, Priyanka Timsina, Archana Shrestha.

**Formal analysis:** Shristi Thapa.

**Methodology:** Shristi Thapa, Priyanka Timsina, Archana Shrestha.

**Supervision:** Shristi Thapa, Priyanka Timsina, Bandana Paneru, Archana Shrestha.

**Writing – original draft:** Shristi Thapa.

**Writing – review & editing:** Shristi Thapa, Bandana Paneru, Archana Pokhrel.

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
