## [Decision Letter · Decision Letter 0]

19 Sep 2025

PGPH-D-25-01465

Human Papillomavirus (HPV) Vaccine Uptake and its Associated Factors among Adolescent Girls in Kathmandu District

Dear Dr. Thapa,

Thank you for submitting your manuscript to PLOS Global Public Health. After careful consideration, we feel that it has merit but does not fully meet PLOS Global Public Health’s publication criteria as it currently stands. Therefore, we invite you to submit a revised version of the manuscript that addresses the points raised during the review process.

We look forward to receiving your revised manuscript.

Kind regards,

Nebiyu Dereje, MPH, PhD

Academic Editor

Journal Requirements:

1. Please clarify all sources of funding (financial or material support) for your study. List the grants (with grant number) or organizations (with url) that supported your study, including funding received from your institution.

2. State the initials, alongside each funding source, of each author to receive each grant.

3. State what role the funders took in the study. If the funders had no role in your study, please state: “The funders had no role in study design, data collection and analysis, decision to publish, or preparation of the manuscript.”

4. If any authors received a salary from any of your funders, please state which authors and which funders.

2. Please send a completed 'Competing Interests' statement, including any COIs declared by your co-authors. If you have no competing interests to declare, please state "The authors have declared that no competing interests exist". Otherwise please declare all competing interests beginning with the statement "I have read the journal's policy and the authors of this manuscript have the following competing interests:"

3. In the online submission form, you indicated that Data is available upon request.

3. Uploaded as supplementary information.

Additional Editor Comments:

Reviewer #1:

Lines 50-52 need to be corrected or reframed.

Lines 53-55 can be edited to a sentence

Lines 55-57 should be revised

The in-text references should come before the full stop.

The author should be consistent with the use of respondent throughout the manuscript.

Reviewer #2:

The manuscript reports the findings of a study that aims to understand the uptake of HPV vaccine among the school-going adolescent girls in Kathmandu district and the factors associated with this uptake. With various governments in the process of introducing this vaccine as part of their national immunisation programmes, this study seems to contribute to the understanding of the factors associated with the uptake of the same.

The authors have described the methods and analysis approach in detail. Clarification on the ethical considerations especially because the study involves children are also made clear. Data is not made available but the authors have responded to the PLOS question on data availability that they would be shared upon request.

One key consideration for the authors to further clarify in their manuscript is regarding the conceptualisation of vaccine uptake. To make it clear as to how is the process or phenomenon or issue of vaccine acceptance or uptake being viewed in this study? This will be useful for the authors to then substantiate the choice of the factors (or lack of) in the study. In the process of describing this, the authors then can explicitly mention the factors that couldn't be explored due to the nature of the conceptualisation itself for eg: access to the vaccine, agency of the child and parents to name a few.

Reviewers' comments:

Reviewer's Responses to Questions

**Comments to the Author**

1. Does this manuscript meet PLOS Global Public Health’s publication criteria?

Reviewer #1: Yes

Reviewer #2: Yes

2. Has the statistical analysis been performed appropriately and rigorously?

Reviewer #1: Yes

Reviewer #2: Yes

3. Have the authors made all data underlying the findings in their manuscript fully available (please refer to the Data Availability Statement at the start of the manuscript PDF file)?

Reviewer #1: Yes

Reviewer #2: No

4. Is the manuscript presented in an intelligible fashion and written in standard English?

Reviewer #1: No

Reviewer #2: Yes

Reviewer #1: Lines 50-52 need to be corrected or reframed.

Lines 53-55 can be edited to a sentence

Lines 55-57 should be revised

The in-text references should come before the full stop.

The author should be consistent with the use of respondent throughout the manuscript.

Reviewer #2: The manuscript reports the findings of a study that aims to understand the uptake of HPV vaccine among the school-going adolescent girls in Kathmandu district and the factors associated with this uptake. With various governments in the process of introducing this vaccine as part of their national immunisation programmes, this study seems to contribute to the understanding of the factors associated with the uptake of the same.

The authors have described the methods and analysis approach in detail. Clarification on the ethical considerations especially because the study involves children are also made clear. Data is not made available but the authors have responded to the PLOS question on data availability that they would be shared upon request.

One key consideration for the authors to further clarify in their manuscript is regarding the conceptualisation of vaccine uptake. To make it clear as to how is the process or phenomenon or issue of vaccine acceptance or uptake being viewed in this study? This will be useful for the authors to then substantiate the choice of the factors (or lack of) in the study. In the process of describing this, the authors then can explicitly mention the factors that couldn't be explored due to the nature of the conceptualisation itself for eg: access to the vaccine, agency of the child and parents to name a few.

**Do you want your identity to be public for this peer review?** For information about this choice, including consent withdrawal, please see our Privacy Policy

Reviewer #1: No

Reviewer #2: **Yes:** Swathi S Balachandra

---

## [Decision Letter · Decision Letter 1]

11 Nov 2025

PGPH-D-25-01465R1

Human Papillomavirus (HPV) Vaccine Uptake and its Associated Factors among Adolescent Girls in Kathmandu District

Dear Dr. Thapa,

Thank you for submitting your manuscript to PLOS Global Public Health. After careful consideration, we feel that it has merit but does not fully meet PLOS Global Public Health’s publication criteria as it currently stands. Therefore, we invite you to submit a revised version of the manuscript that addresses the points raised during the review process.

We look forward to receiving your revised manuscript.

Kind regards,

Nebiyu Dereje, MPH, PhD

Academic Editor

Journal Requirements:

Additional Editor Comments (if provided):

Please address the comments made by Reviewer 3.

Reviewers' comments:

Reviewer's Responses to Questions

**Comments to the Author**

Reviewer #2: All comments have been addressed

Reviewer #3: All comments have been addressed

publication criteria?

Reviewer #2: Yes

Reviewer #3: (No Response)

3. Has the statistical analysis been performed appropriately and rigorously?

Reviewer #2: Yes

Reviewer #3: (No Response)

4. Have the authors made all data underlying the findings in their manuscript fully available (please refer to the Data Availability Statement at the start of the manuscript PDF file)?

Reviewer #2: (No Response)

Reviewer #3: Yes

5. Is the manuscript presented in an intelligible fashion and written in standard English?

Reviewer #2: Yes

Reviewer #3: Yes

Reviewer #2: (No Response)

Reviewer #3: Review Reports

Title: Human Papillomavirus (HPV) Vaccine Uptake and its Associated Factors among Adolescent Girls in Kathmandu District

Manuscript ID: PGPH-D-25-01465R1

Review Comments

General Comments

The standard of the journal should be used consistently

The formatting needs change

Use of present tense E.g. In line 89 this study aims?

Sections fail to contain what it should scientifically contain.

Specific Comments

On the scope

Is that post HPV vaccination survey or not?

If Vaccination is given for them, why you intended to study to it?

After administration of the vaccination, could we study uptake or hesitancy?

Do the eight municipalities part of the Kathmandu district?

On the Title

Add year and possibly the study design to the title

On the Abstract

Incomplete objective E.g. Add Nepal after the district

In line 13-14 you stated that cervical cancer is the most common cancer among women.... How most common is that?

The objective can be merged to the background and be part of the background at the end of the paragraph

In line 22 The participants completed a self-administered survey. Needs rewriting.

The methods section is incomplete.

The prevalence among 14-15-year-old girls were 12.74%. is doubtable. Where are the other adolescents? If they enter to school at age 7 their current age at grade nine and ten will be 16 and 17. Therefore what are you saying?

The result should contain Confidence interval for the prevalence.

The result is inconsistent E.g. Why you use the APR, CI and the P value interchangeably.

Inconsistency in use of words E.g. 1.88 times,45% lower, more likely

The conclusion is weak and it lacks recommendation.

The key words are incomplete

Introduction

Regional magnitude is missing

The definition, causative agent, severity, associated factors and efforts made globally and regionally as well as nationally were missed.

The knowledge gap is widely missed

Methods

The study setting is incomplete E.g. How many health facilities and what type of services are available in the setting?

The study design is not written

The source and the study population were not correctly written

The inclusion criteria are not firm.

Is cluster appropriate sampling procedure for this study?

If you have sampling frame why you used systematic sampling?

What is the source of the tool, why used structured and what does that mean? What was the duration of the question [time frame]?

Rather than pre-test, what was done to ensure quality? What type of validity and reliability?

The ethics is incomplete

Results

It should be written well. E.g. For instance when do we say most?

Tables are not self-explanatory E.g. Table 2.

The results section can be widen more than this

Check for the statistics

Differentiate which variables are associated in binary and multiple logistic Poisson regression?

Discussion and the consequent sections

The discussion is weak

Some of the explanations lack reference E.g. In line 335-39These low rates 335 across Nepal, Uganda, and Nigeria reflect shared challenges, including limited awareness, misinformation, cultural stigma surrounding reproductive health, and logistical issues such as inconsistent vaccine supply. Many cited a lack of information about vaccine availability as a key reason for not getting vaccinated, findings that align with studies from Ethiopia and Uganda”

It should entail appropriate explanation with necessary references employing both theoretical and practical implications

Avoid some of the sentences E.g. The focus of the government of Nepal on adolescents aged 14 and 15 years [you can put it in the introduction section].

Why you acknowledge the department?

What is the role of other authors?

Is there conflict of interest?

Regards,

**Do you want your identity to be public for this peer review?** For information about this choice, including consent withdrawal, please see our Privacy Policy

Reviewer #2: **Yes:** Swathi S Balachandra

Reviewer #3: No

---

## [Editor Report · Decision Letter 2]

13 Jan 2026

Human Papillomavirus (HPV) Vaccine Uptake and Its Associated Factors among Adolescent Girls in Kathmandu District, Nepal: A Cross-Sectional Study

PGPH-D-25-01465R2

Dear Ms Thapa,

We are pleased to inform you that your manuscript 'Human Papillomavirus (HPV) Vaccine Uptake and Its Associated Factors among Adolescent Girls in Kathmandu District, Nepal: A Cross-Sectional Study' has been provisionally accepted for publication in PLOS Global Public Health.

Best regards,

Nebiyu Dereje, MPH, PhD

Academic Editor